# MC-Injection Molding with Liquid Silicone Rubber (LSR) and Acrylonitrile Butadiene Styrene (ABS) for Medical Technology

**DOI:** 10.3390/polym15193972

**Published:** 2023-10-02

**Authors:** Mohammad Ali Nikousaleh, Ralf-Urs Giesen, Hans-Peter Heim, Michael Hartung

**Affiliations:** Institute of Material Engineering, Polymer Engineering, University of Kassel, 34125 Kassel, Germany

**Keywords:** LSR, ABS, multicomponent injection molding, aging

## Abstract

The multicomponent injection molding of liquid silicone rubbers (LSR) with thermoplastics, such as polybutylene terephthalate (PBT) or polyamide (PA), is a state-of-the-art technique and is used in the manufacturing process for many components in the automotive industry and in the field of sanitary engineering. Standard thermoplastics, such as acrylonitrile butadiene styrene (ABS), cannot be bonded with silicone rubbers in injection molding because of their low heat deflection temperature. In this study, we investigated ABS grades approved for medical applications to show how dynamic mold heating and various pretreatment methods for thermoplastic surfaces can be used to produce ABS-LSR test specimens. In addition, such components’ sterilization effect on the adhesive bond will be shown.

## 1. Introduction

Due to their unique properties, silicone elastomers, particularly liquid silicone rubbers, are highly suitable for medical technology applications [1]. They can function across a broad temperature range, are well tolerated by the body, and undergo various sterilization methods. Using established assembly processes, these elastomers are incorporated into medical components like seals, membranes, or valves within thermoplastic housings. Presently, substituting this approach with a multicomponent injection molding technique, which could enhance production efficiency, automation, and quality, faces technical limitations due to the notable differences in processing and mold temperatures of standard thermoplastics. Nevertheless, this obstacle can be overcome by implementing dynamic temperature control in LSR cavities [2]. Although this method has been tested on some thermoplastic–LSR combinations, it offers many potential pairings, especially for thermoplastics with low heat resistance. While the heat deflection temperature for ABS averages around 95–105 °C, the LSR mold temperature ranges from 150 to 230 °C [3,4].

The adhesion between thermoplastics and LSR is a decisive feature for the quality of the subsequent product, which, in addition to the injection molding process, depends above all on the material pairing, surface activation, and subsequent conditioning of the overall component [5,6]. Surface activation in the medical environment can be achieved using shortwave ultraviolet (UVC), plasma, or pyrosil treatment. For the surface treatment of ABS, UVC treatment has proven to be effective concerning peel resistance [7]. Furthermore, an analysis of surface activation following the diverse activation methods was performed using Fourier transform infrared spectroscopy (FTIR spectroscopy) by employing an FTIR spectrometer (Shimadzu, IRAffinity-1S, Kyoto, Japan). This endeavor aimed to understand the reasons for adhesion to the LSR material. The above processes are suitable for cleanrooms and can be integrated into an existing injection molding cycle. Given the demands of medical technology, the polymers used in commercial products must endure a variety of specialized conditions. Among these, the sterilization process stands out, occurring in different variations such as ethylene oxide (EO) or gamma sterilization [8]. In this pursuit, a range of application-oriented post-treatment processes and aging mechanisms were meticulously selected to evaluate the adhesive durability of the employed material combinations under diverse environmental influences. The primary objective of these post-treatments is to comprehensively understand the extent to which material combinations and pretreatment methods can influence adhesive properties, potentially resulting in the integration of a stable multicomponent component.

## 2. Experimental

### 2.1. Injection Molding Process and the Production of Test Specimens/Parts

For the production of the test specimens, the fully electric multicomponent injection molding machine Allrounder 370A 600-70/70 from ARBURG GmbH and Co., KG (Lossburg, Germany), with a maximum clamping force of 600 kN, was used. It has two horizontal injection units arranged in the L position. Here, the LSR unit with a screw diameter of 18 mm and a cooled needle valve nozzle was set in the central axis. The material of the LSR was supplied via a cartridge with the ratio of 1:1. The thermoplastic unit with a screw diameter of 22 mm was placed on the minor axis. The used mold had two cavities: a water-tempered cavity for thermoplastics and an LSR cavity that could be dynamically tempered and into which the thermoplastic hard component was inserted. With the use of dynamic temperature control, we aimed to establish a process in which the LSR is sufficiently vulcanized in the shortest possible time, and simultaneously, the test specimen is cooled together with the thermoplastic to a demolding temperature so that the component can be demolded without deformation. The CycleTemp^®^ Vario type P180M temperature control unit from Wenz Kunststoff GmbH and Co., KG (Lüdenscheid, Germany), was used for dynamic temperature control. Water was used as the heat transfer medium. The temperature control unit allows for a maximum flow temperature of 180 °C and conveys the water at a maximum rate of 60 L/min. A control signal from the injection molding machine can switch between cold water and hot water circuits. Temperature control channels close to the contour (spacing 2 mm), in conjunction with the high flow rate, enable the mold inserts to become hot or cold quickly. In addition, heat transfer depends on the thermal conductivity of the steel.

In order to produce good and reproducible adhesion between LSR and ABS, surface treatment is required. In UVC irradiation, the ABS surface was irradiated with a UVC irradiation module integrated into the process on the handling robot during transfer to the LSR cavity for the 20 s with a low-pressure mercury vapor lamp designated U9W-45ozon from Dinies Technologies GmbH. An Openair^®^ plasma system from Plasmatreat GmbH (Steinhagen, Germany) was used for the surface treatment with plasma. The rotating plasma nozzle was moved externally over the ABS substrate with a robot at a distance of 18 mm and 0.05 m/s speed. Moreover, in this study, we chose an activation speed of 0.4 m/s and a flame distance of 20 mm for surface treatment using pyrosil (Figure 1). These parameters were selected to promote the applied particles’ optimal and dense layer growth.

During the pretreatment with high-energy UVC radiation, polymer chains are broken, and additional photo and ring oxidations occur. This results in the formation of additional functional groups such as hydroxyl, carbonyl, or carboxyl groups. Another effect is that free radicals and ozone from ambient air additionally interact with the polymer surface [9,10].

Atmospheric plasma treatment also initiates chain scissions and produces carbonyl and carboxyl groups via oxidation. In addition, imide or urethane groups can be generated, which increases the surface energy [11].

Pyrosil was produced through the flame pyrolytic deposition of amorphous silicate (SiO_2_). Gas cartridges containing a propane–butane mixture with activation components were used. When exposed to a high-temperature flame (1300 °C), the cartridge contents is converted into silicate particles, which settle as ash. This process forms a highly reactive, moisture-stable silicon oxide (SiOx) layer. By incorporating reactive groups, the surface energy can be significantly increased [12].

### 2.2. Artificial Aging

When considering the adhesion between materials, the aging of subsequent components is often neglected or not considered. In the tests carried out here, the components were therefore subjected to artificial aging. In the first step, the composites were stored in a forced-air oven at 70 °C for 24 h. In the second step, the influence of humidity was considered by storing the specimens for seven days at 70 °C and 95% humidity and for 14 days at 70 °C and 95% humidity in a climatic chamber. The selected parameters reflect the maximum possible application ranges.

### 2.3. Sterilization Process

In medical technology, the three most commonly used processes are superheated steam sterilization, sterilization with ethylene oxide, and gamma sterilization. The hard–soft composites used here were sterilized with gamma irradiation. The sterilization plant of B. Braun Melsungen AG (Melsungen, Germany) was used for this purpose. This type of sterilization is a cold sterilization process using electromagnetic irradiation. Due to low heat input, and because it is carried out without the use of chemicals, the process is well suited for thermoplastics with low heat resistance. The used specimens were exposed to a dose of 25 kGy. The disadvantages of the treatment can be thermoplastic discoloration and hardening. The optical and mechanical properties can thus change. For silicone rubbers, it is known that a so-called self-healing phenomenon can occur, as a result of which slits reconnect and therefore cannot be opened. Furthermore, it is known that some silicone rubbers become slightly harder [13].

In ethylene oxide sterilization, medical devices are treated with the highly toxic gas ethylene oxide, through which microorganisms are killed. This process is particularly suitable for materials that are neither thermally nor steam-stable. The temperatures in this process are between 37 and 63 °C with humidity between 40% and 80% [14]. In addition, this process is also used in thermoplastics, as it is known that thermoplastics, such as PP, are not radiation-resistant. In the investigations presented here, a process with a temperature of about 55 °C and a humidity of 30% was chosen. The components were sterilized at ALMO-Erzeugnisse Erwin Busch GmbH (Bad Arolsen, Germany).

### 2.4. Materials

For the tests presented here, we used materials that have medical device approval or are certified for these applications according to BFR and FDA. The selected ABS grades are primarily used for disposable medical devices, such as infusion systems or elastomer pumps, because of their good mechanical properties and relatively low prices. The silicone rubber used in this study is suitable for hot water and drinking water applications.

#### 2.4.1. Thermoplastics

Terluran^®^ GP 22 from INEOS Styrolution Group GmbH (Frankfurt am Main, Germany) was selected as a versatile ABS. It is used for household appliances, automotive construction, toys, and medical technology. It is primarily processed using injection molding. The material has high impact strength and good surface quality, and it is very easy to color. Its heat deflection temperature (HDT-A) is between 94 °C and 99 °C. The tensile strength is 45 MPa, the modulus of elasticity is 2300 MPa. Its processing temperatures range from 220 °C to 260 °C.

The second material used for the tests was Elix ABS M203FC from ELIX Polymers SLU (Tarragona, Spain). This material is specially developed for medical technology and is primarily used for respirators, auto-injection devices, and medical housings. Special properties include good flow ability and excellent gloss values. Its mechanical properties are in a similar range to the properties of the other ABS used. The tensile strength is 46 MPa, and the modulus of elasticity is 2400 MPa. Its datasheet shows that the heat deflection temperature (HDT-A) is between 94 °C and 98 °C. The processing temperature for injection molding is 240 °C.

The third material used for the tests was Lustran 348 resin, an injection molding grade of ABS with medium impact strength and good tensile properties. It is especially suitable for medical applications manufactured by the INEOS Styrolution Group GmbH. Special properties include good flow ability and high gloss values and use for components of intravenous systems, surgical instruments, and diagnostic test kits. The tensile strength is 52 MPa, and the modulus of elasticity is 2600 MPa. According to its datasheet, the heat deflection temperature (HDT-A) is approximately 96°. The processing temperature for injection molding ranges between 230 °C and 260 °C.

#### 2.4.2. Silicone Rubber

An Elastosil 3271/45 from Wacker Chemie AG (Munich, Germany) was used for the tests. This material is a two-component self-adhesive silicone rubber suitable for injection molding. The mixing ratio of components A and B is 1:1. It has been primarily developed for applications of hard–soft compounds in drinking water and hot water systems, and therefore, it has drinking water approval. The Shore A hardness is 45, the tensile strength is 6.3 MPa, and the elongation at break is 540%.

### 2.5. Test Methods

#### 2.5.1. Fourier Transform Infrared (FTIR) Spectroscopy

The chemical compositions of unmodified and activated surfaces of the ABS materials were investigated using FTIR spectroscopy (Shimadzu, IRAffinity-1S, Kyoto, Japan). The FTIR spectra were obtained through optical contact at room temperature within the standard wavenumber range of 4000 to 600 cm^−1^, using a crystal (ZnSe at 45°) with a higher refractive index than the sample. The wavenumber resolution of 4 cm^−1^ was used for recording.

#### 2.5.2. Peeling Test according to VDI 2019

The bond strength of the manufactured test specimens was investigated utilizing peel tests according to VDI guideline 2019. This guideline defines a procedure for testing hard–soft composites’ bond strength or peel resistance. The peel resistance W_s_ and the fracture pattern serve as characteristic values for the test. The peel resistance is the quotient of the average peel force and the width of the common contact surface of the soft and hard components transverse to the peel direction. The peeling tests within the scope of this work were carried out on the universal testing machine Inspekt table blue from the company Hegewald and Peschke Mess- und Prüftechnik GmbH (Nossen, Germany). The testing machine has a maximum nominal load of 5 kN and a maximum crosshead travel of 1265 mm. According to the VDI 2019 guideline, the LSR strip is peeled off from the TP substrate at an angle of approximately 90°. The peeling speed is 100 mm/min (Figure 2).

## 3. Results and Discussion

### 3.1. FTIR Spectroscopy

The FTIR spectra of ABS were primarily distinguished by the nitrile C≡N signal at 2237 cm^−1^ and the two peaks corresponding to the aromatic C-H stretches at 758 and 698 cm^−1^. The aromatic C-H stretches were observed above 3000 cm^−1^, and the aliphatic C-H stretches were located just below 2922 and 2851 cm^−1^ [11,15,16,17]. Subsequent experiments were conducted to understand better changes in the chemical structure of the ABS substrate using FTIR spectroscopy results before and after surface treatments.

#### 3.1.1. FTIR Analysis of Samples after UVC Irradiation

The FTIR spectra of ABS before and after UVC irradiation are presented in Figure 3. Following the FTIR analysis of the ABS after UVC irradiation, an increase was observed in the hydroxyl regions (OH-stretch) and the carbonyl C=O bands within the range of 3600–3200 cm^−1^ and 1850–1650 cm^−1^, respectively. Simultaneously, a decrease in the region corresponding to the two unsaturated bonds, trans-1,4 (967 cm^−1^) and vinyl-1,2 (911 cm^−1^) C=C, was evident as the UVC irradiation time increased [15,16,17]. The nitrile, aromatic, and aliphatic groups appeared to be less impacted due to the higher binding energy of the C≡N bond, which exceeds the energy of UVC irradiation [18].

#### 3.1.2. FTIR Analysis of Samples after Openair^®^ Plasma Treatment

Figure 4 presents the FTIR spectra of ABS before and after atmospheric plasma treatment. Consistent with existing studies, the plasma treatment of ABS reduces the region under the band attributed to the C=C double bonds of styrene (within the range of 1600 to 1400 cm^−1^). Specifically, the band’s breadth at 1450 cm^−1^ varies as treatment time increases, which is attributed to the oxidation of the C=C double bond. Moreover, emerging peaks at 1730 and 1280 cm^−1^ signal the emergence of polar C-O groups within the aliphatic chain. These groups could potentially be linked to carboxylic acid groups, as evidenced by the emergence of bands at 1320 and 1210 cm^−1^, corresponding to the C-O group [11,19]. As shown in the figure, the difference in the FTIR spectroscopy of the surface after the extended plasma treatment is not very noticeable.

#### 3.1.3. FTIR Analysis of Samples after Pyrosil Treatment

Figure 5 illustrates the outcomes of FTIR spectroscopy, comparing non-activated and activated ABS samples. To demonstrate the activation process through FTIR analysis more effectively, the surfaces were subjected to activation using pyrosil, with 2 and 10 rounds of treatment. A complete conversion of the precursor to SiO_2_ occurred, as indicated by stretching vibrations within the range of 1250–1000 cm^−1^. Furthermore, additional Si-C bonds were potentially present within the 1090–1030 cm^−1^ range. Existing research also suggests the possible strengthening of Si-NH2 bonds in the 1250–1100 cm^−1^ range [12,20,21].

### 3.2. Peeling Test

The results of the peel tests show the adhesion between ABS and LSR due to the surface treatments and the influence of different aging and sterilization processes. Without pretreatment, the material combination exhibited no adhesion. Figure 6 shows the results with Openair^®^ plasma. Without post-treatment, the average peel resistance rates for all thermoplastics were 2.5 to 3.3 N/mm. After storage at 70 °C for 24 h, adhesion decreased by 20% to 30%. After storage at a temperature of 70 °C and a relative humidity of 95% for seven days, the adhesion collapsed by up to 90%. This may indicate that moisture diffuses through the LSR into the interface and hydrolytically degrades the bonds responsible for adhesion [22]. Another factor, particularly in higher humidity conditions, is the degradation of the crosslinker within silicone components during artificial aging. This degradation becomes evident as the crosslinker depletes rapidly in the initial days of aging [23,24].

The results of pyrosil pretreatment are shown in Figure 7. The average peel resistance rates without post-treatment for all thermoplastics were the same as the rates observed in plasma pretreatment, in the range of 2.6 to 3 N/mm. The greater peel resistance before post-treatment indicates that the pyrosil pretreatment leads to heightened surface energy and facilitates chemical bonding with the LSR component [17,25]. After storage at 70 °C for 24 h in the same process, adhesion decreased by 20% to 30. The results after storage at a temperature of 70 °C and a relative humidity of 95% for 7 and 14 days also show a significant decrease. Still, the losses are not as pronounced as those observed with plasma treatment. This suggests that the pretreatment method can affect the hydrolytic degradation of the bonds [22]. Here, the degradation of the crosslinker within silicone components during artificial aging also contributes to the reduction in adhesive stiffness [23,24].

The peel resistance rates for UVC-pretreated composites were similar (Figure 8). The adhesion values decreased after 24 h of storage at 70 °C without humidity influence. No advantages over plasma treatment were observed during this aging process. After the maximum conditioning of 7 and 14 days, the peel resistance rates decreased by more than 50%. Interestingly, however, higher adhesion was evident here, compared with pretreatment with the pyrosil and Openair^®^ plasma. Similar to other activation methods, the degradation of the crosslinker in silicone components during artificial aging also plays a role in diminishing adhesive properties [23,24]. Additionally, it could be postulated that the radicals generated from UVC irradiation form more robust bonds with the silicone components, resulting in more stable properties against temperature and humidity [18].

As a result of sterilization, the values for both thermoplastics also dropped (Figure 9). EO sterilization exhibited the least influence, with a drop of approx. 25% for both thermoplastics. Gamma sterilization was more damaging to the adhesion. The adhesion loss in this case was up to 80% for the components. These results indicate that gamma electromagnetic radiation cleaves or dissolves adhesion or the chemical or physical bonds responsible for adhesion formation. In addition, gamma radiation had a negative effect on the mechanical properties of materials and led to the failure of the LSR component. The reduction in EO sterilization may be due to the high gas permeability of LSR. Although EO sterilization is only carried out on the surface, good gas permeability can cause the EO gas to penetrate the interface and have a damaging effect.

Figure 10 displays the results of the peel resistance test conducted on ABS-LSR with pyrosil pretreatment after undergoing various sterilization processes. The findings indicate that sterilization has an adverse impact on adhesion properties. Still, the effect was less noticeable than plasma pretreatment. In contrast, as a result of the treatment with gamma sterilization and a combination of gamma sterilization and aging, adhesion remained above 1 N/mm, corresponding to approximately one-third of the original bonding.

Post-treatment with ethylene oxide also showed an adhesion-reducing effect for UVC-pretreated composites of Terluran^®^ GP-22 and Elix ABS M203FC (Figure 11). Without post-treatment, the average peel resistance rates for all thermoplastics were above 3 N/mm. A higher standard deviation could be observed for the samples after sterilization. Here, a drop of about 25% on average was also observed for the thermoplastics. In contrast to the Openair^®^ plasma and pyrosil surface pretreatment, some adhesion could be observed for Terluran^®^ GP-22, Elix ABS M203FC, and Lustran^®^ 348 with UVC irradiation pretreatment in conjunction with gamma sterilization with downstream artificial aging.

Regardless of the activation methods, after treatment with ethylene oxide, the peel resistance decreased by approximately 25%. The systematically higher standard deviation in the EO sterilized samples can be attributed to the sterilization process and the possibly uneven gas penetration. The changed material properties of the pure silicone component can also explain this decrease. This manifests itself via increased cohesive failure. As shown in studies by Gautriaud et al. [12], the ethylene oxide treatment of LSR can lead to a decrease in tensile strength. However, the influence on the mechanical properties is otherwise insignificant. This may explain the sometimes premature cohesive cracking and the higher standard deviation observed in this study. Conversely, gamma irradiation significantly influences the mechanical properties of silicone rubbers depending on the irradiation dose. The chain scission and formation of radicals occur, leading to a decrease in tensile strength, tensile elongation, tensile strength, and elasticity [8]. This can explain the accumulation of premature cohesive cracks. In conclusion, the type of surface activation of thermoplastics for hard–soft composites with silicone rubber with the required post-treatment processes, especially for medical products, must be decided depending on the change in mechanical, chemical, and adhesion properties. This concerns the chemical and mechanical properties of materials so that the products can maintain their functionality until the end of their service life.

## 4. Conclusions and Outlook

The three factors considered, namely material pairing, activation process, and subsequent component conditioning, demonstrated that no general assumptions or predictions can be made about the adhesion of the joining partners. In particular, the material pairing factor strongly influences subsequent adhesion and does not reveal any statistically significant tendency. For medical technology or industries with similarly high requirements, the MK injection molding process for standard thermoplastics may be of interest. However, it was shown in this study that post-treatments often have a significant effect on service life and function. Since a large number of material combinations have not yet been investigated, this factor can be assumed to have the most significant potential for increasing adhesion. For components subject to only moderate mechanical requirements or offering a sufficiently large adhesion surface, pretreatment using UVC, pyrosil, or plasma, which can be easily integrated into cleanroom production, can already be used today. Furthermore, prefixation during the assembly process via adhesion, which is then supported by an additional force-fit component, is also conceivable. In this way, the components are positioned automatically but can also be subjected to the highest forces in the adhesion area. The form-fit delineation of the adhesion through precise surface activation allows for the movement of components such as diaphragms, valves, or telescopic functions. The range of potential components and their automation potential make it interesting to continue research in this direction in the future.

The research and development project “MeKoMed” is funded by the German Federal Ministry of Education and Research (BMBF) in the program “Innovations for Tomorrow’s Production, Services and Work” (funding code 02P18C054) and supervised by the Project Management Agency Karlsruhe (PTKA). The authors are responsible for the content of this publication.

## Figures and Tables

**Figure 1 polymers-15-03972-f001:**
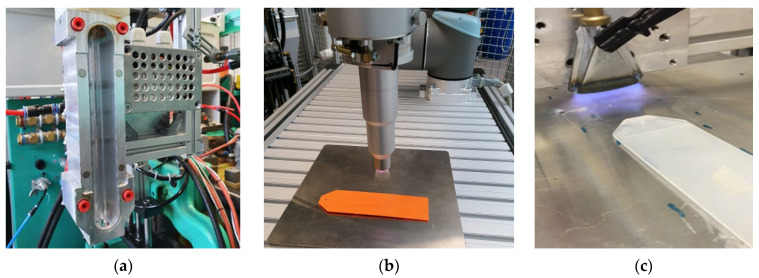
The UVC lamp installed on the robot (**a**), the Openair^®^ plasma process (**b**), and the pyrosil process (**c**).

**Figure 2 polymers-15-03972-f002:**
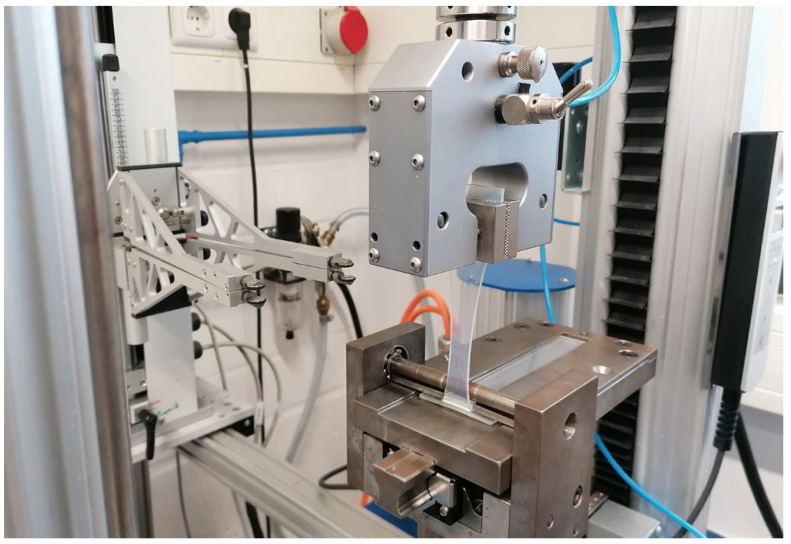
Peel test according to VDI 2019.

**Figure 3 polymers-15-03972-f003:**
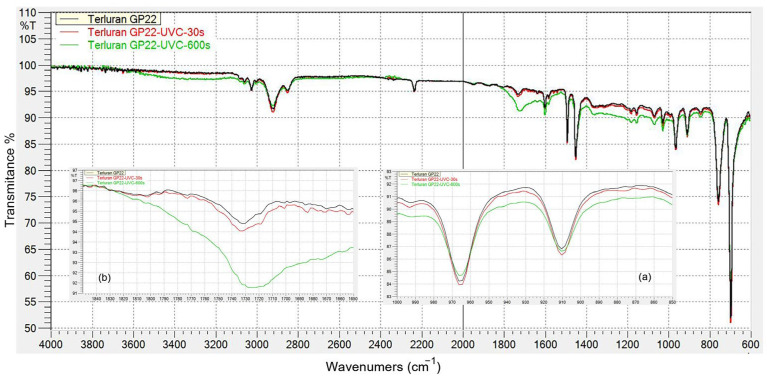
FTIR spectra of ABS surface before and after UVC irradiation: (**a**) butadiene region; (**b**) carbonyl region.

**Figure 4 polymers-15-03972-f004:**
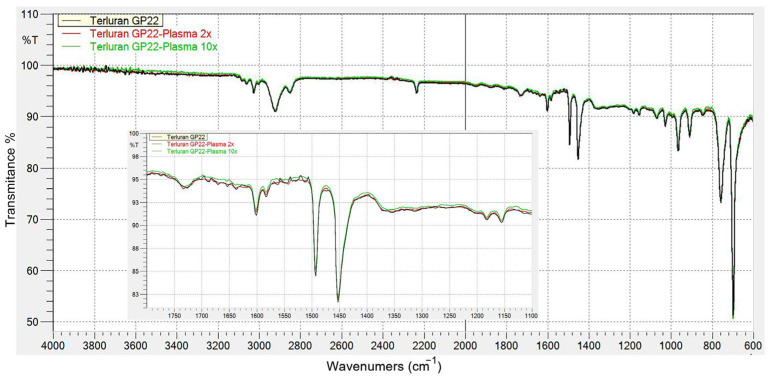
FTIR spectra of ABS surface before and after 2 and 10 rounds of Openair^®^ plasma surface treatment.

**Figure 5 polymers-15-03972-f005:**
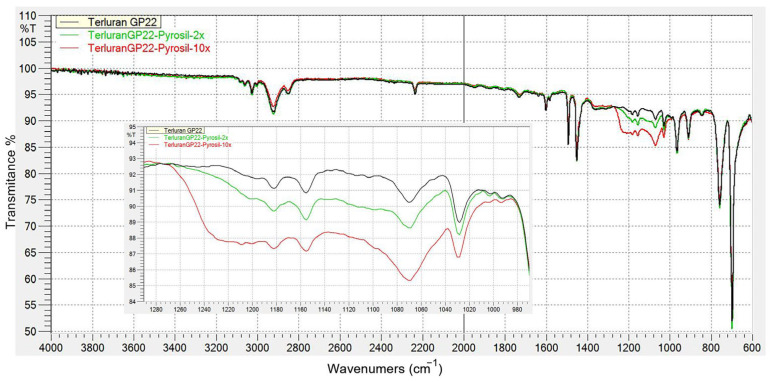
FTIR spectra of ABS surface before and after before and after 2 and 10 rounds of pyrosil surface treatment.

**Figure 6 polymers-15-03972-f006:**
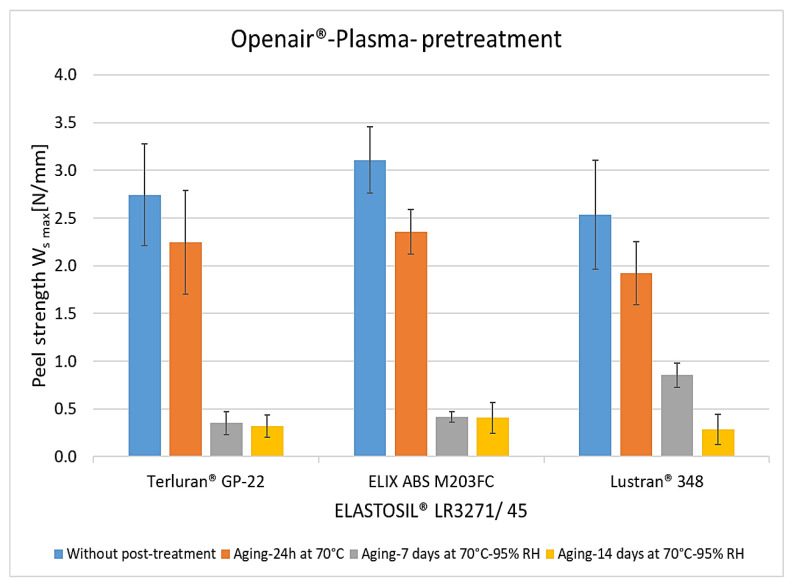
The peel resistance of ABS-LSR with Openair^®^ plasma pretreatment and different post-treatments (aging for 24 h at 70 °C, aging for 7 and 14 days at 70 °C, and 95% RLF).

**Figure 7 polymers-15-03972-f007:**
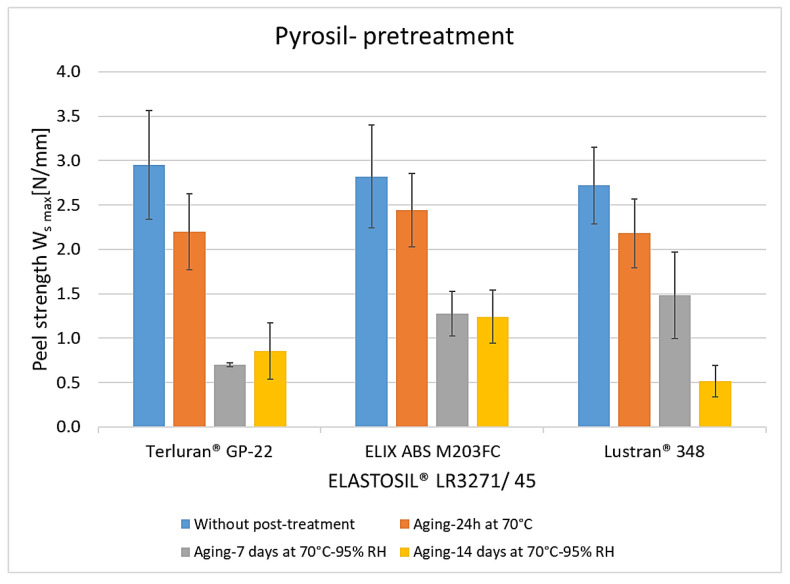
The peel resistance of ABS-LSR with pyrosil pretreatment and different post-treatments (aging for 24 h at 70 °C, aging for 7 and 14 days at 70 °C, and 95% RLF).

**Figure 8 polymers-15-03972-f008:**
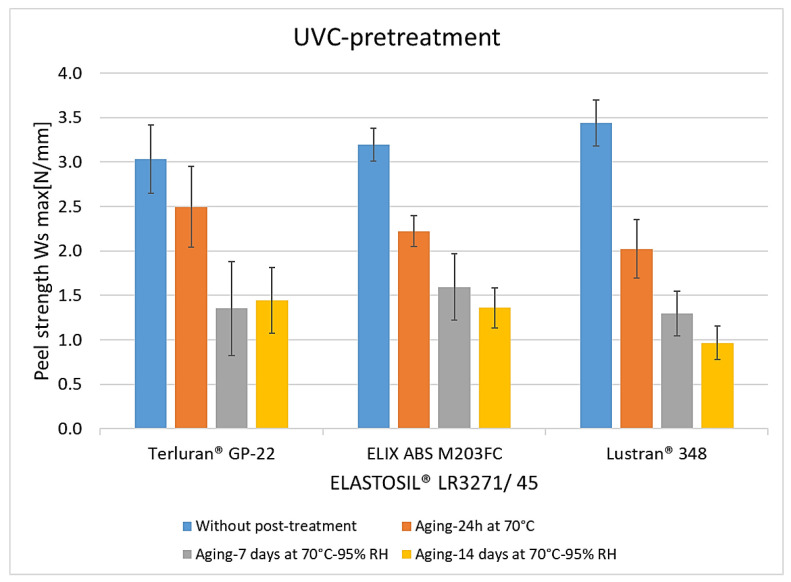
The peel resistance of ABS-LSR with UVC pretreatment and different post-treatments (aging for 24 h at 70 °C, aging for 7 and 14 days at 70 °C, and 95% RH).

**Figure 9 polymers-15-03972-f009:**
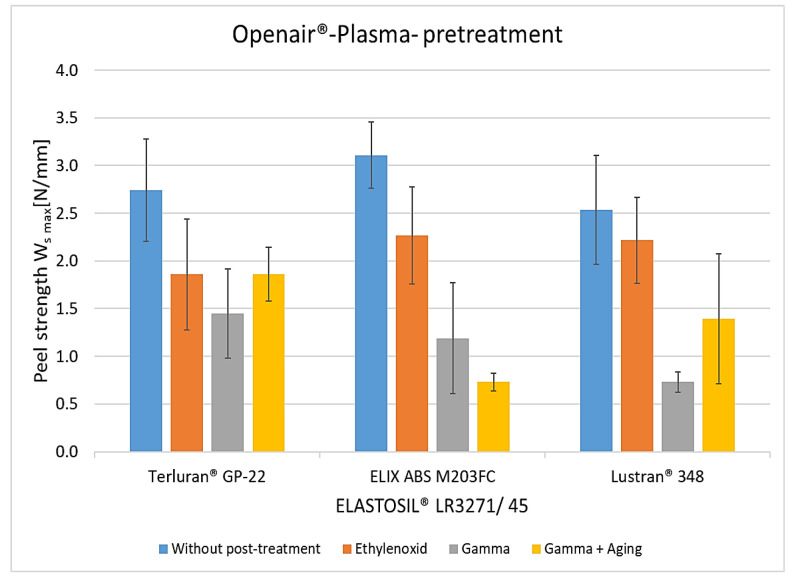
The peel resistance of ABS-LSR with Openair^®^ plasma pretreatment and different sterilization methods.

**Figure 10 polymers-15-03972-f010:**
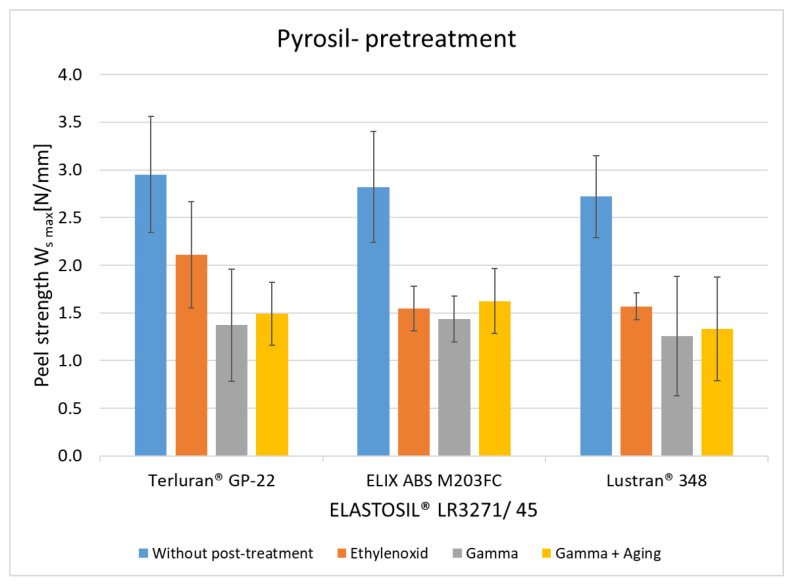
The peel resistance of ABS-LSR with pyrosil pretreatment and different sterilization methods.

**Figure 11 polymers-15-03972-f011:**
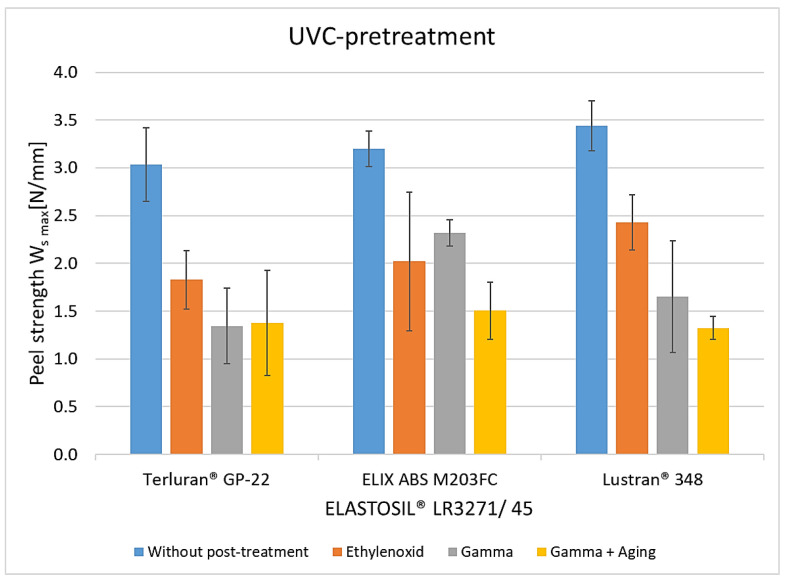
The peel resistance of ABS-LSR with UVC pretreatment and different sterilization methods.

## Data Availability

Not applicable.

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
