# Peer review of "MC-Injection Molding with Liquid Silicone Rubber (LSR) and Acrylonitrile Butadiene Styrene (ABS) for Medical Technology"

_polymers, 2023, doi:10.3390/polym15193972_

Round 1

Reviewer 1 Report

In this manuscript studies MC-Injection Molding with Liquid Silicone Rubber (LSR) and 2 Acrylonitrile Butadiene Styrene (ABS) for Medical Technology.

For medical technology with similarly requirements, the MC-Injection Molding process for standard thermoplastics can be of interest.  Furthermorte it was shown that post-treatments often have a significant effect on service life and function.

The subject is interesting and the manuscript has been well written. The results have been well presented and discussed. 

Author Response

no comments to do

Reviewer 2 Report

In the paper, an issue of multi component injection molding with liquid silicone rubber and acrylonitrile butadiene styrene for medical technology has been presented.

This is an interesting and valuable study on the important problem of polymer injection molding. The paper is clearly written and well documented. The conclusions are clear and logical. The paper has got a high practical value.

However, the paper resembles a laboratory study report. It would be worth pointing out and developing the scientific elements of the conducted research.

English is OK

Author Response

Editing of English is done

the methods are better described

Reviewer 3 Report

The manuscript entitled "MC-Injection Molding with Liquid Silicone Rubber (LSR) and Acrylonitrile Butadiene Styrene (ABS) for Medical Technology" deals with the characterization of the adhesion between ABS and a silicone rubber, in specimens obtained through multi-component injection molding.

In my view, the topic of the manuscript looks interesting and deserves investigation. However, the manuscript is poor prepared, the obtained results are merely presented without a deep investigation or a critical comparison with results obtained for similar systems. Therefore, I suggest to the Authors to revise the manuscript, also taking into account the following comments:

- Please, explicitate the acronyms when appear for the first time in the text (as an example, PBT in the abstract).

- the introduction part is quite poor. Please, improve it by adding some references on relevant works on the same topic. Furthermore, in the final part, please introduce the aim of the work and the main obtained results.

- the obtained results need to be critically commented. Actually, in the manuscript only a mere description of the Peel resistance data for the different systems is reported. Please, enhance the scientific soundness of the manuscript.

Author Response

  • Introduction is revised, more literature dicussion is made.
  • acronyms are described
  • Description of the results were revised in a clear way 

Round 2

Reviewer 2 Report

Thank you for the improvement of the paper.

Reviewer 3 Report

The manuscript can be accepted forpulicatin, as the Authors revised it according to the suggestions of the Reviewers.